# Comprehensive mapping of lunar surface chemistry by adding Chang'e-5 samples with deep learning

Chen Yang [1,2] ✉, Xinmei Zhang [1], Lorenzo Bruzzone [3], Bin Liu [2], Dawei Liu[2], Xin Ren [2], Jon Atli Benediktsson [4], Yanchun Liang [5], Bo Yang[5], Minghao Yin[6], Haishi Zhao [5] ✉, Renchu Guan [5] ✉, Chunlai Li [2] ✉ & Ziyuan Ouyang[2,7]

Lunar surface chemistry is essential for revealing petrological characteristics to understand the evolution of the Moon. Existing chemistry mapping from Apollo and Luna returned samples could only calibrate chemical features before 3.0 Gyr, missing the critical late period of the Moon. Here we present major oxides chemistry maps by adding distinctive 2.0 Gyr Chang'e-5 lunar soil samples in combination with a deep learning-based inversion model. The inferred chemical contents are more precise than the Lunar Prospector Gamma-Ray Spectrometer (GRS) maps and are closest to returned samples abundances compared to existing literature. The verification of in situ measurement data acquired by Chang'e 3 and Chang'e 4 lunar rover demonstrated that Chang'e-5 samples are indispensable ground truth in mapping lunar surface chemistry. From these maps, young mare basalt units are determined which can be potential sites in future sample return mission to constrain the late lunar magmatic and thermal history.

The surface of the Moon is the critical interface between the lower boundary of the lunar atmosphere and the upper boundary of the crust. It is a window through that we can view the composition of the crust and the history of the Moon[1]. The entire lunar surface is covered with a layer of lunar regolith, except on some very steep-sided crater walls and lava channels. Remote sensing technology, i.e., the high-energy or optical techniques, such as gamma ray, neutron spectroscopy (GRNS) and X-ray spectroscopy[2], and missions, i.e., Clementine[3–5], SMART-1[6], Chang'e-1[7–9], Chandrayaan-1[10], Lunar Reconnaissance Orbiter (LRO) missions[11] and SELENE (KAGUYA)[12], are important measurements of the chemical properties of lunar material from lunar regolith. It is critical that the surface composition inferred by remote sensing is calibrated with the actual abundances. Lunar samples collected by six Apollo and three Luna missions from lunar regolith provide the valuable ground truths and have resulted in a significantly enhanced

understanding of the global distributions of the lunar surface chemistry. However, the samples in the low latitude region returned by the Apollo and Luna missions only revealed the evolution of Moon 3.0 Gyr ago, missing the critical late period of the Moon[13]. Young lunar soil samples with different chemical characteristics are necessary for a more accurate surface chemistry estimation.

On 1 December 2020, China's first lunar-sample-return mission Chang'e-5 (CE-5) successfully landed in the northeastern Oceanus Procellarum (51.916°W and 43.058°N), which is one of the youngest mare basalt units[14]. The Chang'e-5 sampling region explored a high latitude that was not reached by the previous Apollo and Luna sampling missions. After ten months, the age of the Chang'e-5 sample, i.e., 2.030 ± 0.004 Gyr ago (Ga) was reported by the Institute of Geology and Geophysics, Chinese Academy of Sciences[15]. Meanwhile, the study of the Key Laboratory of Lunar and Deep Space Exploration, National

[1]College of Earth Sciences, Jilin University, Changchun, China. [2]Key Laboratory of Lunar and Deep Space Exploration, National Astronomical Observatories, Chinese Academy of Sciences, Beijing, China. [3]Department of Information Engineering and Computer Science, University of Trento, Trento, Italy. [4]Faculty of Electrical and Computer Engineering, University of Iceland, 102, Reykjavik, Iceland. [5]College of Computer Science and Technology, Jilin University, Changchun, China. [6]College of Information Science and Technology, Northeast Normal University, Changchun, China. [7]Institute of Geochemistry, Chinese Academy of Sciences, Guiyang, China. ✉e-mail: yangc616@jlu.edu.cn; zhaohs@jlu.edu.cn; guanrenchu@jlu.edu.cn; licl@nao.cas.cn

Astronomical Observatories, Chinese Academy of Sciences shows that the minerals and compositions of Chang'e-5 soils are consistent with mare basalts and can be classified as low-Ti/low-Al/low-K type with lower rare-earth-element contents than materials rich in potassium, rare earth element and phosphorus[16]. Chang'e-5 soils have high FeO and low Mg index, which could represent a new class of basalt[16]. It is proven that Chang'e-5 samples carry information about young volcanic activity on the Moon and are an indispensable ground truth in mapping lunar surface chemistry[17].

Owing to the high-resolution advantage of the optical technology compared with high-energy technology, existing research mostly uses optical images to estimate the abundance and distribution of major oxides on lunar surface. Abundance algorithms and inversion models are two main methods for the surface compositional inferences. The abundance algorithms determined only the $TiO_2$ and FeO contents with the defined sensitive parameters by performing a coordinate rotation in UV/VIS ratio versus VIS reflectance space[5,18]. For the inversion models, traditional linear and nonlinear regression methods, such as the partial least squares regression[19,20], neural network[9,21–23] and support vector machine[24], have applied to establish the relationships between the spectral values and ground truths of major oxide (e.g., $TiO_2$, FeO, $Al_2O_3$, MgO, CaO, and $SiO_2$) abundances at the lunar sampling sites. Throughout existing global distribution maps, varying degrees of differences are present in chemical abundance even with the same lunar samples[11].

At present, three unavoidable factors, i.e., the limited number of lunar samples, the resolution of optical images and the complex relationship between the spectral characteristics and oxide contents, make estimation results uncertain. (i) a wide gap exists between the number of sampling points and the amount of values to be estimated. For example, the number of sampling points of the Clementine, Chang'e-1 and SELENE for spectral parameters are 47[5], 18[8], and 53[18], and for inversion models are 19[21], 39[9], and 40[23]/38[24], but over billions oxide abundance values should be calculated. (ii) every Apollo mission includes some closer distance sampling points in which both the geologically uniform and the inhomogeneous soils produced by mixtures of local mare materials with highland ejecta are present[11]. The returned samples collected in these areas may be not necessarily fully representative of the surface observed from orbit. Remote sensing pixels measure areas dozens to thousands of meters in size, whereas the returned samples are from areas that are typically much smaller than a square meter. (iii) the spectral albedo characteristics and values show complicated relations between the oxide abundances and their relationships. It is difficult to establish these complicated relationships by the traditional inversion models. Thus, estimating reliable and accurate lunar surface chemical contents is the basic for extending subsequent scientific research.

The Chang'e-5 samples considered in this research were analyzed by X-ray fluorescence spectrometer (XRF) (sample CE5C0800YJF-M002)[16]. The major elements were analyzed (Na, Mg, Al, K, Ca, Ti, Fe, and Mn). The SELENE (KAGUYA) multiband imager (MI) data, characterized by the abundant spectral features (UV–vis and NIR spectroscopy with eight wavelengths) and a high spatial resolution (59 m/pixel)[25], are selected to infer abundances of six major oxides ($TiO_2$, FeO, $Al_2O_3$, MgO, CaO, and $SiO_2$). The MI reflectance data have been terrain shade-corrected. Meanwhile, the number of sampling points were expanded to 55 according to the high spatial resolution of MI data (Supplementary Table 1). Furthermore, a deep learning (DL) algorithm, i.e., a 1D convolutional neural network[26], was designed to establish an oxide inversion models and to acquire more accurate lunar surface chemical contents.

## Results and discussion
### Maps of lunar surface chemical abundances
For mapping lunar surface chemical abundances, a deep learning-based inversion method with a 1D convolutional neural network[26] (Methods) was designed to model the complex relationships between the six major oxide abundances and MI values with the Apollo, Luna and Chang'e-5 sampling points. Due to the small sample size, the leave-one-out cross-validation (LOOCV) was adopted for evaluating the generalization ability of the inversion method and avoiding overfitting and underfitting[27]. In this work, the MI spectral features and the measured oxide abundances from sample-return sites both by adding Chang'e-5 samples and by only with Apollo and Luna data (w/o Chang'e-5) have been used to build two training sets to derive the inversion model. The average test precision is used as the validation accuracy of the inversion model. After training and testing, the abundances inversion models of six major oxides, i.e., $TiO_2$, FeO, $Al_2O_3$, MgO, CaO, and $SiO_2$ were established (Methods). The prediction accuracies of the inversion model and the verification accuracies of LOOCV for six major oxides are shown in Supplementary Fig. 1. The average and the best determination coefficients ($R^2$) for the prediction of oxide abundances are all greater than 0.99 with both training sets. The $R^2$ values for the validation of oxide contents are all greater than 0.90 (Supplementary Fig. 2). The high precision value confirms that the established inversion models achieve reliable performance on all six major oxides.

The inversion models between the MI spectral features and the measured oxide abundances, where derived two kinds of lunar surface chemistry maps, one with only Apollo and Luna samples (w/o Chang'e-5) (Supplementary Fig. 2) and the other adding with Chang'e-5 samples (Fig. 1). The average abundances of $TiO_2$, FeO, $Al_2O_3$, MgO, CaO, and $SiO_2$ when adding Chang'e-5 samples are 1.2 wt%, 8.94 wt%, 19.9 wt%, 12.62 wt%, 9.23 wt%, and 45.05 wt%; w/o Chang'e-5 are 1.23 wt%, 8.59 wt%, 22.02 wt%, 14.24 wt%, 8.3 wt%, and 45.13 wt%. From a global perspective, the major six oxides in both cases with Chang'e-5 and w/o Chang'e-5 maps exhibit trichotomy distributions. The maria show more $TiO_2$, FeO and MgO and the highlands have higher $Al_2O_3$, CaO and $SiO_2$ abundances. The oxides in the SPA basin are different from the maria and highlands, $TiO_2$ and FeO are lower than those on mare surfaces, and $Al_2O_3$ and CaO are in between those in the maria and the highlands, while MgO and $SiO_2$ are higher than those on highlands. However, these results point out significant differences of oxide abundances between cases with Chang'e-5 and w/o Chang'e-5. In the maria, the Chang'e-5 has various degrees higher $TiO_2$, FeO, $Al_2O_3$, CaO and $SiO_2$ abundances with respect to the w/o Chang'e-5 except for the MgO. On highland surfaces, the Chang'e-5 results in lower $Al_2O_3$ and CaO abundances and higher MgO values compared with the w/o Chang'e-5. In the SPA basin, the Chang'e-5 displays higher $Al_2O_3$ and CaO and lower MgO and $SiO_2$ abundances. The differences of six major oxides $TiO_2$, FeO, $Al_2O_3$, MgO, CaO, and $SiO_2$, between the measured abundances of Chang'e-5 samples (Supplementary Table 1) and w/o Chang'e-5 are 3.85 wt%, 5.44 wt%, −0.73 wt%, −0.09 wt%, −2.58 wt% and 2 wt%, respectively. Obviously, the inversion results present significant errors when using only Apollo and Luna data.

Two no-sample returned lunar landing site regions, i.e., Chang'e-3 and Chang'e-4, are selected to demonstrate the validity of the Chang'e-5 inversion results (Table 1). For Chang'e-3, the latest corrected in situ chemical compositions have been acquired by Active Particle-induced X-ray Spectrometer (APXS) on Yutu rover[28]. Compared to the results acquired by the APXS, the chemical contents of $TiO_2$ and FeO in the inversion model by adding the Chang'e-5 samples are respectively 0.73 wt% and 0.03 wt%, whereas those of $Al_2O_3$, MgO, CaO, and $SiO_2$ are 5.3 wt%, 2.15 wt%, 2.28 wt% and 3.37 wt% higher. However, the chemical contents of $TiO_2$ and FeO in the w/o Chang'e-5 inversion model have distinct differences relative to the APXS abundances, especially the $TiO_2$ and FeO values down to 3.5 wt% and 3.23 wt%, respectively. This may be due to the fact that Chang'e-3 and Chang'e-5 are both on young Eratosthenian lava flows[29–32] and the values of $TiO_2$ and FeO are close to each other (Table 1 and Supplementary Table 1). Chang'e-5 samples represent a new type of differentiated lunar basaltic rock compared with Apollo and Luna samples, which have distinctive

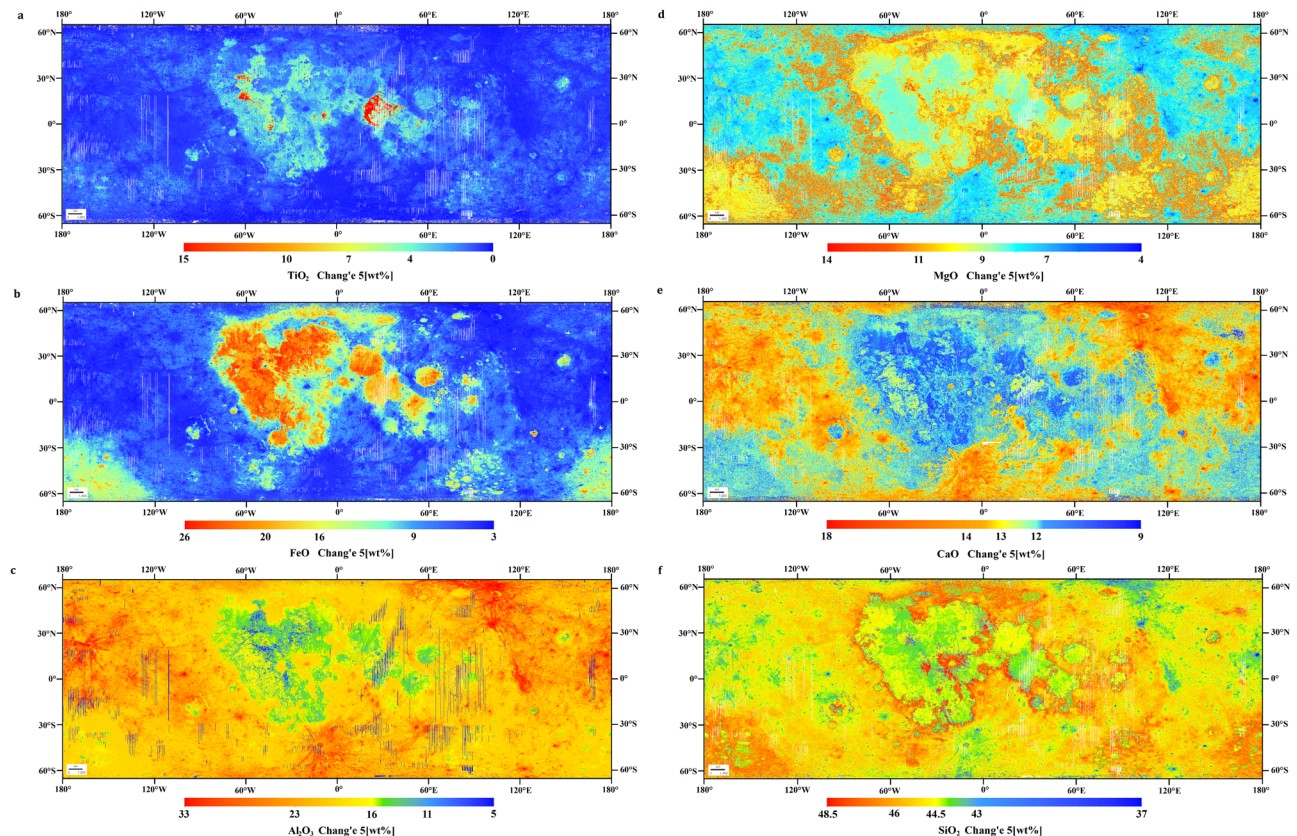

**Fig. 1 | The lunar surface chemistry maps of six major oxides by including the Chang'e-5 samples. a–c** shows the maps of $TiO_2$, FeO, and $Al_2O_3$ abundances respectively calculated from the deep learning-based inversion method. **d, e** and **f** show the maps of MgO, CaO and $SiO_2$ abundances respectively calculated from the deep learning-based inversion method.

**Table 1 | Chemical compositions of the Chang'e-3 and Chang'e-4 landing sites**

|  | Chang'e 3 |  |  | Chang'e-4 |  |  |  |
| --- | --- | --- | --- | --- | --- | --- | --- |
|  | Chang'e-3 Yutu APXS | Chang'e-5 | w/o Chang'e-5 |  | Chang'e-4 LPR | Chang'e-5 | w/o Chang'e-5 |
| $TiO_2$(wt%) | 3.88 | 3.15 | 0.38 | %(FeO+$TiO_2$) | 9±4%<br>11±4% | 14.66 | 14.70 |
| FeO(wt%) | 22.6 | 22.57 | 19.37 |  |  |  |  |
| $Al_2O_3$(wt%) | 10.83 | 16.13 | 14.75 |  |  |  |  |
| CaO(wt%) | 9.93 | 12.21 | 12.24 |  |  |  |  |
| MgO(wt%) | 10.57 | 8.42 | 8.68 |  |  |  |  |
| $SiO_2$(wt%) | 41 | 44.37 | 41.57 |  |  |  |  |

The abundances of six major oxides ($TiO_2$, FeO, $Al_2O_3$, CaO, MgO, and $SiO_2$) analyzed with Chang'e-3 Yutu APXS (Active Particle-induced X-ray Spectrometer) and (FeO + $TiO_2$) contents found by the Chang'e-4 LPR (Lunar Penetrating Radar) are from refs. 29, 34, respectively.

$TiO_2$ (5 wt%) and significantly higher FeO content (22.5 wt%). Most mare basalts from Apollo and Luna collections have $TiO_2$ contents in the range 0.6–10 wt%, and abundances between 5 and 6 wt% are relatively lacking. Furthermore, the FeO contents are all below 21 wt% in the Apollo and Luna samples. Therefore, the w/o Chang'e-5 inversion model results in a large error in the $TiO_2$ and FeO contents. For Chang'e-4, the inferred (FeO + $TiO_2$) contents are 14.66 wt% and 14.70 wt% in the Chang'e-5 and w/o Chang'e-5 models, there is a good agreement with the range 9 ± 4% and 11 ± 4% found by Chang'e-4 LPR at this site[33]. In general, the abundances of six major oxides inferred by adding Chang'e-5 samples exhibit more reasonable values.

### Comparison with previous lunar chemical maps

We compared the inversed lunar surface chemistry results with the Lunar Prospector (LP) Gamma-Ray Spectrometer (GRS) maps[34] (Supplementary Fig. 3), and the estimated contents by Clementine UVVIS[5],

Diviner CF[22], Chang'e-1 IIM[9], and SELENE MI data[23,24] with abundance algorithms or inversion methods (Supplementary Fig. 4), respectively. The oxide abundances on the global or lunar lithologic unit's surfaces[35] between the inversed map and the previous maps show varying degrees of difference.

In the comparison of LP GRS maps, only $TiO_2$, FeO, and $Al_2O_3$ in the lunar maria were selected for the low precision of the MgO, CaO, and $SiO_2$ abundances determined from the LP GRS data. To accurately simulate the spatial response function of the LP GRS that sampled at 2°per pixel (61 km pixel scale at the equator), a Gaussian weight function[11] is used. The correlation between the LP GRS and SELENE MI abundance maps with Chang'e-5 was examined by extracting the average and standard deviation of each mare (Supplementary Fig. 3). For $TiO_2$ and FeO, the comparison of LP GRS maps and SELENE MI abundance maps with Chang'e-5 average values for each mare fall near the 1:1 line, while $R^2$ are 0.623 and 0.633, respectively. Meanwhile, $TiO_2$

and FeO contents in LP GRS and SELENE MI with Chang'e-5 abundance maps both are comparable with the most lunar samples points (Supplementary Fig. 4). It indicates that LP GRS maps and SELENE MI abundance maps with Chang'e-5 show broadly consistent estimates in TiO$_2$ and FeO contents. However, obvious deviation exists between Al$_2$O$_3$ LP GRS and SELENE MI abundance maps. The SELENE MI abundance maps compared with Chang'e-5 maps shows higher Al$_2$O$_3$ abundances (+1–6 wt%) in most lunar maria, particularly in Oceanus Procellarum and Mare Imbrium. It should be noted that Al$_2$O$_3$ LP GRS contents are generally lower than in lunar samples points (Supplementary Fig. 4). In terms of the rich spectral features and high spatial resolution, the SELENE MI abundance maps are much better than Al$_2$O$_3$ contents estimated.

The Chang'e-5 inversion accuracy has been further evaluated by comparing with the Clementine UVVIS[5], Chang'e-1 IIM[9], Diviner CF[22], and SELENE MI data[23,24] between measured abundances and the estimated values. The measured abundances are calculated from the sample compositional data and estimated values are extracted from oxide prediction result (Supplementary Fig. 4). As shown in Supplementary Fig. 4, Chang'e-5 inversion model achieved the best performance using the high spatial resolution SELENE MI data with the most lunar samples points. The Diviner CF have a large error due to their relatively low resolution. In contrast, the three SELENE MI results have small differences in the six major oxides contents. It is worth noting that the estimated values of Chang'e-5 inversion model have good consistency with the measured abundances at almost all the sampling sites, especially the Chang'e-5 landing site.

The root mean square errors (RMSEs) (Methods, Supplementary Table 2) show that the Chang'e-5 inferred values of the major oxides are the closest to the measured abundances. The RMSEs of Chang'e-5 inferred six major oxides are significantly smaller than the RMSEs of the Clementine UVVIS, LP GRS and Diviner CF oxides. The RMSE of Chang'e-5 FeO is only 0.0027 higher than the Chang'e-1 IIM; other oxides prediction accuracy is better than the Chang'e-1 IIM. For the SELENE MI, the inversion models, i.e., Zhang et al.[23] and this work by adding with Chang'e-5 samples shows obvious good performance. However, the RMSEs of oxides indicated that results of this work are more accurate, the errors of TiO$_2$, FeO, Al$_2$O$_3$, MgO and CaO derived in this paper are smaller than those in Zhang et al.[23], i.e., 0.0449, 0.1556, 0.1479, 0.0939, 0.2555 and 1.10, respectively. There are two main reasons for this. In this work, we selected all the returned samples abundances for mapping lunar surface chemistry. Note that the in situ chemical contents of Chang'e-3 data are also used in Zhang et al.[23]. The in situ chemical contents of Chang'e-3 data are derived based on APXS measurements that exist deviation between lunar samples[28,36], which may introduce uncertainty in the process of inversion. On the other hand, the inversion model of this work adopted the convolution layer with a convolution kernel size of 1 × 3, which can model the interaction information among long-range spectra more effectively than the convolution kernel size of 1 × 2 used in Zhang et al.[23]. Meanwhile, the convolution operation with the stride of 2 is used in the second and fourth convolution layers to achieve the goal of downsampling and enlarging the receptive field of the model. Therefore, this work can reduce the information loss in the process of spectral information extraction when model the relationship between the spectra and the oxides abundances.

It should be emphasized that the estimated oxide in existing lunar chemical maps with abundance algorithms and inversion models only represents the upper-most surface materials[11]. Meanwhile, estimated abundances can only reflect the regional average values of the lunar surface materials for the complex geologic settings. For example, the range of SiO$_2$ contents in lunar chemical maps based on Chang'e-1 IIM, Diviner CF, and SELENE MI with Chang'e-5 samples are about 39–48.2 wt%, 37.3–46.4 wt%, and 37–48.5 wt%, respectively. The silicic regions on the Moon, i.e., Hansteen Alpha, Gruithuisen Domes,

Aristarchus Crater, Lassell Massif, Helmet, Apennine Bench, Mairan Domes, and Compton-Belkovich volcanic complex, the SiO$_2$ contents in the inversed map are 45.09 wt%, 44.36 wt%, 43.14 wt% (crater rim is about 46%), 47.36 wt%, 46.59 wt%, 46.82 wt%, 46.98 wt% and 42.87 wt%, respectively. The estimated SiO$_2$ contents in these silicic regions are relatively high compared with the other lunar surfaces. However, most of these silicic regions are located at the interchange of the lunar mare and highlands or in the lunar mare, such as Aristarchus craters where the Si-rich materials are confined to the rim and ejecta[37–39]. Higher resolution orbital images from future spacecraft missions, particularly returned samples from the silicic regions, will allow for examination of silicic materials from the upper subsurface, which in many cases could reveal the original surface SiO$_2$ abundances before mare contamination.

## Lunar surface chemistry in the division of geologic units

The molar or atomic ratio of Mg/(Mg+Fe) symbolized as Mg# reflects the ratio of Mg to Fe in rocks or minerals and is related to the source region, composition, and partial melting degree of the original magma and the magmatic evolutionary progress[40]. The Mg# map for the Moon calculated with the Chang'e-5 inferred MgO and FeO is shown in Fig. 2a. The average Mg# value across the Moon is 0.53, which is close to the Clementine Mg# 0.57[40] optimized with gamma-ray spectroscopy data, and lower than the LP GRS Mg # 0.606[35], Diviner Mg # 0.652[22], Chang'e-1 IIM-derived Mg # 0.646[9], and SELENE MI Mg # reported 0.675[23] and 0.67[24]. Meanwhile, the separability of Chang'e-5 inferred Mg # in lunar maria (0.4), highlands (0.58), and South Pole-Aitken basin (0.46) are prominent. The division of three lunar geologic units, i.e., the lunar maria (LM), Feldspathic Highland (FH), and South Pole-Aitken (SPA) units are refined (Fig. 2a). The extensive high-Mg# regions are more highlighted at the centre of feldspathic highland terrain, i.e., Freundlich Sharonov, Dirichlet Jackson basins, Compton-Bel'kovich, Milikan, Chappell-Debye, and Fowler-Klute, also around the Apollo 16 site, compared with the results from the Kaguya Spectral Profiler[41]. Therefore, the Chang'e 5 samples play an important role in revealing mineral and petrologic characteristics of the Moon and in re-establishing the lunar magma ocean (LMO) model.

In the lunar maria (LM), a young mare basalt unit is determined based on inversed inferred compositions, as shown in Fig. 2b. They are mainly distributed in the centre of the Procellarum-KREEP-Terrane[42], with two content division, i.e., low-medium TiO$_2$ (4–6 wt%), high FeO (>18 wt%), low Al$_2$O$_3$ (<15 wt%) abundance and high TiO$_2$ (>6 wt%), FeO (>18 wt%), Al$_2$O$_3$ (>15 wt%) abundance. In the first content range, Chang'e-5 is a typical representative; whereas, no returned samples are available in the second range. For further exploring the late lunar magmatic and thermal history with Chang'e-5 lunar samples, compositional units were defined according to the distribution of oxide abundances. In the definition of lunar geologic units, compositional homogeneity is an essential precondition[43]. Each compositional unit is formed within a short period and each unit represents a single volcanic eruptive phase providing consistency in oxide abundances. It is very important to define compositional units to obtain reliable age determinations for analyzing the eruption of lunar young mare basalts. Twenty-six young mare basalts compositional units (U1-U26) were defined and mapped in this study. Each compositional unit has distinct element composition, including low-median-Ti and high-Ti mare basalts[44,45] that reflect complex magmatism. The model ages of young mare basalts were referred to Hiesinger et al.[46] and Qian et al.[47]. Two distinct young mare basalts regions were identified, i.e., the basalts located in the southwest of Kepler crater (U12) and that below the Aristarchus Plateau crater (U7). U12 has the highest TiO$_2$ (6.69 ± 1.4 wt%), also high FeO (20.15 ± 1.02 wt%), and Al$_2$O$_3$ (16.23 ± 0.96 wt%) abundances with the model age (1.3–1.4 Ga[46,47]) that is different from the abundance of existing returned samples. U7 has

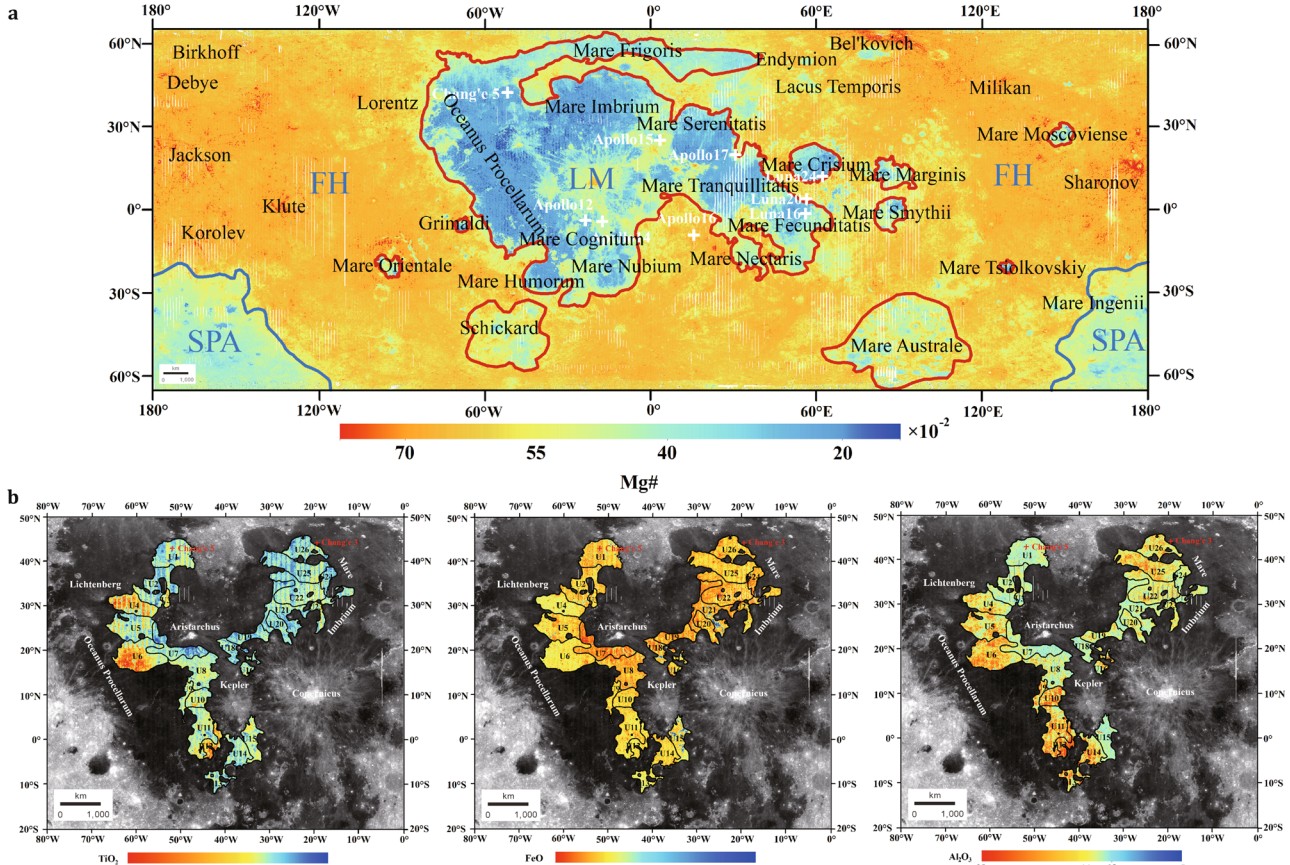

**Fig. 2 | The partition of geologic units based on the inversed lunar surface chemistry maps. a** Mg# map across the Moon. The Mg # map highlights the boundaries of three lunar geologic units, i.e., LM, FH, and SPA. The red lines encircle the LM unit, the blue lines mark the SPA unit, and the remaining region is the FH unit. **b** Young mare basalts unit determined based on the inferred TiO$_2$, FeO, and Al$_2$O$_3$ compositions. The black lines denote the boundaries of the young mare baslts units. The "+" denotes the Chang'e-5 ang Chang'e-3 landing sites. The age of young mare basalts was from Hiesinger et al.[43] and Yuqi Qian[44].

low TiO$_2$ (3.78 ± 1.1 wt%), high FeO (22.13 ± 1.56 wt%) and low Al$_2$O$_3$ (14.37 ± 0.98 wt%) abundances which is similar to Chang'e 5 samples but with different model ages (1.0 Ga[48] and 1.9 Ga[47]). Therefore, more samples are required, especially in the two distinct young mare basalts units, for understanding the late lunar thermal evolution.

The SPA unit is a compositionally unique region on the Moon, the titanium and iron abundances on the nonmare of this basin range from 0 to 1.5 wt% and from 9 to 16 wt%, which is consistent with an approximate 1:1 mixture of lower crustal material of the Moon and mantle rock containing <0.1wt% TiO$_2$ and 10μ16 wt% FeO[48]. The results indicate that the SPA unit is a mixture of mantle and lower-crust material[48]. Meanwhile, the average MgO abundance and Mg# within SPA unit are 10% ± 0.8% and 0.46, respectively, which are lower than expected for mantle material, suggesting that the materials in SPA unit represent a relatively ferroan composition and mafic complement to ferroan anorthosites[49]. Although the interpretation of the previous research claims to have identified mantle-derived olivine[50], but further to reduce uncertainty will require a sample from the interior of South Pole-Aitken basin. China's Chang'e 6 mission will collect samples from the South Pole-Aitken Basin on the far side of the Moon by 2024[51]. At that time, the lunar surface chemistry would be further refined on the SPA unit.

## Methods
### Sample selection
In this work, considering spectral and spatial characteristic of SELENE MI data, a total of 115 lunar soil samples acquired from 55 lunar sampling sites of Apollo, Luna, and Chang'e-5 landing regions (which

deemed to be relatively geologically uniform) were selected to represent the ground truth of chemical abundances (Supplementary Table 1)[16,25,52,53]. For Maria, highlands and transitional regions, 9, 19, 42, 1, 6, and 1 samples were selected at Apollo 12, Apollo 15, Apollo 17, luna 16, luna 24, and Chang'e-5 sites, respectively; 31 and 1 samples were selected at Apollo 16 and Luna 20 sites, respectively; and 5 samples were selected at Apollo 14 site, respectively. Due to the higher spatial resolution of the Multiband Imager (MI), the number of sampling stations were refined and expanded at Apollo 12 and 14 compared with ref. 52,53 and [25]. The Apollo 11 samples were not considered for data missing in the MI. The abundances of six major oxides, i.e., TiO$_2$, FeO, Al$_2$O$_3$, MgO, CaO, and SiO$_2$ were recalculated using the samples marked by ref. 53 except for Apollo 17 sites where six oxide abundances were from ref. 25, and Chang'e-5 site from ref. 16. The source literatures of the measured abundances were indicated in Supplementary Data 1.

### Data pre-processing
SELENE (KAGUYA) multiband imager (MI) data have rich spectral features and high spatial resolution. MI has five UV–vis spectral bands at 415, 750, 900, 950, and 1001 nm, and four near-infrared spectral bands at 1000, 1050, 1250, and 1550 nm. The spatial resolution is 20 meters in the five visible bands and 62 meters in the four near-infrared bands at the orbital altitude of 100 km[18]. In this work, the MI reflectance data have a resolution of 59 m/pixel after topography shadow correction[54]. Eight wavebands, i.e., 415, 750, 900, 950, 1001, 1050, 1250, and 1550 nm were employed to calculate the abundances of the six major oxides by removing similar bands, i.e., 1000 nm and 1001 nm. The

spectral features of MI and the corresponding lunar samples for each site are shown in Supplementary Table 1. Two spectral angle parameters, i.e., $\theta_{Ti}$ and $\theta_{Fe}$ were also adopted to suppress the impact of optical maturity (i.e., space weathering) on the inversion of $TiO_2$ and FeO abundances[5,55]. In this work, $\theta_{Ti}$ and $\theta_{Fe}$ were calculated from MI data using the following algorithms[25].

$$\theta_{T_i} = \arctan\{[(R_{415}/R_{750}) - 0.208]/(R_{750} - (-0.108))\} \quad (1)$$

$$\theta_{F_e} = \arctan\{[(R_{950}/R_{750}) - 1.250]/(R_{750} - 0.037)\} \quad (2)$$

where, $R_{415}$, $R_{750}$, and $R_{950}$ represent the reflectance values at the wavelengths of 415 nm, 750 nm, and 950 nm in MI data, respectively.

## Deep learning-based inversion model

This work aims to invert the abundance of major oxides from the spectral images by establishing the complicated relations between the oxide abundances and the spectral albedo characteristics. To accurately predict the oxide abundances on the lunar surface, a deep learning-based inversion model was designed. After pre-processing, a pixel $x_i$ on SELENE (KAGUYA) multiband imager (MI) data represents a local lunar surface region with eight spectral observations, i.e., $x_i = [x_{i0}, x_{i1}, ..., x_{i8}]$, which can be regarded as a one-dimensional sequence. It should be noted that the task in this study is to invert abundance of oxides from a 1D spectral sequences, the reflectance characteristics exhibited by oxides in a certain band on the spectral profile can be regarded as local features. The 1D convolutional neural network (1D CNN)[26] with small-window local connectivity property has the ability to extract local features. Meanwhile, the weight-sharing mechanism enables the 1D CNN to have fewer parameters compared to other models such as Multilayer Perceptrons (MLPs). This makes it more effective for our analysis where there is a very small number of returned lunar samples. Accordingly, we designed a 1D convolutional neural network-based inversion algorithm demonstrated by high accuracy and low risk of overfitting. According to the above considerations, the proposed inversion model contains a feature extraction module which is composed of five 1D convolutional blocks and a prediction module with one fully connected layer. Each convolution block in the feature extraction module consists of a 1D convolution layer, a Batch Normlization (BN)[56] layer and the ReLU[57] activation function. The number of channels of the features in the five convolution blocks is set to [64, 64, 128, 128, 256]. The prediction module predicts the six major oxide abundances corresponding to the spectral observation based on the features extracted by feature extraction module. More specifically, we adopted a two-stage strategy to derive an inversion model with good performance and high robustness for abundance estimation of the major oxides on the lunar surface. In the first stage, leave-one-out cross-validation (for a dataset with N samples, which can be viewed as N-fold cross-validation) was employed to obtain the best 1D CNN inversion model of each fold. For each fold, one sample was selected as the validation set, and the remaining samples were adopted to train the model. The training was done with the following settings: the Adam optimizer was used[58–60], learning rate was initialized to 0.01, weight decay was 0.0001, the max number of epochs was 100 and the batch size was set to the number of the training samples. During the training process of each fold, the evaluation results on the validation set were used to select the best model corresponding to this fold, while an early-stopping strategy based on the accuracy results was also used to avoid the problem of overfitting due to continued training. Thus, N models could be obtained in the first stage. Then, in the second stage, inspired by the work of 59 and 60, a simple yet effective strategy was adopted, i.e., averaging the weights of the obtained N models to derive the final model for predicting the oxide abundance

on the lunar surface. These N averaged models are the best models corresponding to each fold in leave-one-out cross-validation. The weight averaging process can be regarded as a model-level ensemble strategy, which ensures that the final model has a good generalization ability. The model is implemented with PyTorch[61] framework on a PC workstation (Intel(R) Xeon(R) Platinum 8352Y CPU @ 2.20GHz with 128 GB of RAM and NVIDIA GeForce RTX 3090 Graphics Processing Unit).

### Evaluation and validation

To assess of the performance of the proposed deep learning-based inversion model, the root mean square errors (RMSE) and the determination coefficients ($R^2$) are adopted and can be calculated as follows:

$$RMSE = \sqrt{\frac{1}{N}\sum_{k=1}^{N}(y_k - \hat{y}_k)^2} \quad (3)$$

$$R^2 = 1 - \frac{\sum_{k=1}^{N}(y_k - \hat{y}_k)^2}{\sum_{k=1}^{N}(y_k - \bar{y})^2} \quad (4)$$

where $N$ is the total number of samples; $y_k$ and $\bar{y}$ are the oxide abundances of the $k$-th sample and the mean oxide abundances of all samples, respectively; $\hat{y}_k$ represents the estimated oxide abundances of the $k$-th sample by using the inversion model.

In the process of inversion, the leave-one-out cross-validation (LOOCV), an evaluation method for assessing model generalization in the case of small samples, is utilized to validate the effectiveness of the proposed inversion model. Considering a total of $N$ available samples, each sample is selected as a test sample, and the remaining $N$-1 samples are used as the training set to train the inverse model. Then, the trained model estimated the oxide abundances with the test sample. This process is repeated $N$ times to obtain inversion results for $N$ samples, and RMSE and $R^2$ values are considered to evaluate the performance of the inversion model. Lower RMSE and higher $R^2$ values indicate that the inversion model has better performance as well as better generalization ability to avoid overfitting.

## Ablation and comparative experiments

The ablation experiments of the proposed 1D CNN inversion model were performed on the model size and hyperparameters (i.e., learning rate and weight decay). With the same network structure (5 convolution blocks), the model size was changed by adjusting the number of channels in each convolution layer, and four model sizes were examined in the ablation experiments, i.e., 'small': [16, 16, 32, 32, 64], 'medium': [32, 32, 64, 64, 128], 'large': [64, 64, 128, 128, 256], 'huge': [128, 128, 256, 256, 512]. The two hyperparameters, i.e., the learning rate and the weight decay were set to {0.1, 0.01, 0.001, 0.0001} and {0.01, 0.001, 0.0001, 0.00001}, respectively. RMSE and $R^2$ under the LOOCV configuration were used as evaluation metrics. From the results of ablation experiments, one can see that the 'large' 1D CNN inversion model with the learning rate = 0.001 and weight decay = 0.0001 achieved the best LOOCV performance (Supplementary Fig. 5 and Supplementary Data 2).

To demonstrate the superior performance of the proposed 1D CNN inversion model, we compared it with the standard Extreme Learning Machine (ELM)[62] and a new MLPs variant, namely Generalized Operational Perceptrons (GOPs)[63,64]. The latter is capable of using any neuron model, linear or nonlinear, while having a heterogeneous network structure like the human nervous system. The RMSE and $R^2$ under the LOOCV configuration were used to evaluate the performance of each method. From the results of comparative experiments on the six oxides, it can be seen that the 1D CNN inversion model designed on the basis of the considered task and

data characteristics achieved the best LOOCV performance (Supplementary Table 3).

### Comparison strategy

Two kinds of comparison strategies were adopted to demonstrate the importance and the necessity of Chang'e-5 samples when mapping the surface chemistry. In the first strategy, we computed the six major oxide abundances by both using the proposed deep learning-based inversion model with the MI spectral features and measured contents from sample-return sites by adding Chang'e-5 data and only with Apollo and Luna data (w/o Chang'e-5). Please note that the same parameters were used for the network training in the Chang'e-5 and without Chang'e-5 inversion models. Then the difference between the results obtained by adding Chang'e-5 and without Chang'e-5 were analyzed. Meanwhile, the chemical compositions of in situ measurements obtained by Active Particle-induced X-ray Spectrometer (APXS) on the Chang'e-3 Yutu rover[28] and Chang'e-4 Lunar Penetrating Radar (LPR) were also used for quantitative comparison. On the second comparison strategy, the six major oxides abundances investigated by the gamma ray spectroscopy (GRS) data with the Lunar Prospector (LP)[34] and estimated by the Clementine UVVIS[5], Chang'e-1 IIM[9], Driviner CF[22] and SELENE MI[23,24] were introduced for further assessment. In addition, the alteration of division of geologic and compositional units in the Chang'e-5 landing area, i.e., the northeastern Oceanus Procellarum (41–45°N, 49–69°W), was shown.

### Data availability

The maps data generated in this study have been deposited in Figshare [https://doi.org/10.6084/m9.figshare.24081438 and https://doi.org/10.6084/m9.figshare.24460114]. The SELENE (KAGUYA) multiband imager (MI) data with a resolution of 59 m/pixel after topography shadow correction used in this study are available at https://astrogeology.usgs.gov/maps/lunar-kaguya-multiband-imager-mosaics and https://planetarymaps.usgs.gov/mosaic/Lunar_MI_multispectral_maps/, the Clementine UV–vis data are available in https://planetarymaps.usgs.gov/mosaic; and the Lunar Prospector Gamma Ray Spectrometer Elemental Abundance are accessible from https://pds-geosciences.wustl.edu/lunar/lp-l-grs-5-elem-abundance-v1/. The experimental data generated in this study are provided in the Source Data file. Source data are provided in this paper.

### Code availability

The deep learning-based inversion model used in this work is available at https://github.com/hszhaohs/DL-IM.

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

## Acknowledgements

Thanks to all the staff of China's Chang'e-5 Lunar Exploration Project for their hard work on in situ investigation and returning lunar samples. The sample abundances studied in this work were provided by the China National Space Administration. This research was funded by the National Natural Science Foundation of China (Grants Nos. 42272340 and 42241163 to C.Y. and 42302265 to H.S.Z.). We thank Prof. Shengbo Chen for providing global lunar surface chemistry derived from LRO diviner data, and Prof. Xianmin Wang and Prof. Jun Huang for providing the lunar surface chemistry observed by the SELENE (KAGUYA) multiband imager (MI) data.

## Author contributions

C.Y. conceived the research; X.M.Z., H.S.Z. and R.C.G. implemented the deep learning-based inversion algorithm. C.L.L. and Z.Y.O. contributed scientific background, geological interpretation, and consistency of remote-sensing observations. B.L. and X.R. conducted data preprocessing. B. L., D.W.L., J.A.B. and Y.C.L. provided the background knowledge of models and helped to focus on the relevance of the contribution. B.Y. and M.H.Y. contributed to data processing. C.Y., X.M.Z. and H.S.Z. wrote the manuscript.

## Competing interests

The authors declare no competing interests.
