## [Peer Review File · Nature Communications]

Comprehensive mapping of lunar surface chemistry by adding Chang'e-5 samples with deep learningREVIEWER COMMENTS

Reviewer #1 (Remarks to the Author):

Review Comments for the manuscript entitled: “Comprehensive mapping of lunar surface chemistry by adding Chang'e-5 samples with deep learning”

This is an interesting work on Chang'e 5 samples, which were used for training a 6-layer 1D CNN as the inversion model to predict the chemical contents in the new lunar maps precisely. On SELENE (KAGUYA) multiband imager (MI) data, each pixel represents a local lunar surface region with eight spectral observations, and 1D CNN model is thus trained over this 1D data to predict six major oxide abundances corresponding to the spectral observation based on the features extracted by the feature extraction module. The work shows certain merit and improved the earlier results in terms of accuracy; however, I noticed the following issues especially on the implementation of the deep learning (prediction) module, which should be addressed properly on the next revision round.

- 1) The Authors stated that the training data is limited, i.e., “there is a **very small** number of returned lunar samples.” I could not see this number, later on referred as N. What is the size of the train set? This is the reasoning the Authors have mentioned using a 1D CNN model, which in fact has 6 layers and 100s of neurons (channels). If N is really small (e.g., $N < 500$), even this network will easily “overfit”.
- 2) On the other hand, many other network models exist that especially suit to limited training data and have a better learning performance than CNNs, e.g., Transformers, Operational Neural Networks (ONNs), Convolutional Support Estimator Networks (CSEs), etc., especially when considering the CNN model used.
- 3) The Authors used a “Vanilla CNN” model although there are recently much better CNN models with enhanced learning capabilities and superior receptive fields size, e.g., UNet, VGG, ResNet, Inception, etc., providing CNNs with pooling layers, self-attention layers, skip connections, etc. that can improve the performance. Please note that solely using stride=2 will not guarantee an enhanced receptive field size.
- 4) With their limited connection and weight sharing capabilities, CNNs are preferred especially when the data dimension is high, e.g., images with hundreds of pixels or 1D signals with a long data vector like 2048×1 . However, in this work the input 1D data has only a data vector of 8×1 and thus a fully connected network can rather be used such as MLPs or their new variants Generalized Operational Perceptrons (GOPs), or even ELMs. Such networks can learn the interrelations of these 8 elements much better than 1D CNN with only 3×1 kernels.
- 5) In the light of (1-4), the Authors should justify the choice of the network model, configuration and hyper-parameters. An ablation study is a must. In fact they can investigate which network type and model with what configuration and parameters is the best to use in this problem. This will be very valuable.
- 6) The choice of the network model during training is missing. For each fold, the Authors are performing maximum of 100 epochs (and why 100?, did they notice a saturation in the training?) and as far as I see, there is no validation set. So how did you choose the model during training for the evaluation? At the end of the 100th epoch (when training is complete)? If so, how do you

guarantee that this is the model that generalizes the best on the test set? I highly recommend to use a validation set within the train set (e.g, 20% of the train set can be spared for validation set) and then choose the CNN model in a particular epoch that yields the highest accuracy on the validation set. You will surely improve the accuracy further.

- 7) Finally, the Authors should combine deep learning discussion in a single section. Some network details are scattered all along the manuscript. For example, the last paragraph of “**Comparison with previous lunar chemical maps**” section can be moved into the “Deep learning inversion model” section. They can provide all the details and block diagrams for the network models used in the study. This will improve the manuscript significantly.

Reviewer #2 (Remarks to the Author):

Congratulations on the completion of such a large project. I have a handful of minor comments to help improve the flow of the manuscript. I also have a concern about the framing of the SiO₂ map, detailed below. I look forward to reading the completed manuscript – Dr. Sarah Valencia

Abstract

“However, this mapping could only calibrate chemical features before 3.0 Gyr, missing the critical second half of the history of the Moon.” – Saying “half” here implies a much older age than the Moon actually is.

“Therefore, young lunar soil samples with different chemical characteristics are necessary for mapping surface chemistry comprehensively.” – Why is it necessary that the samples are both young and have different chemical characteristics?

Change “by adding 2.0 Gyr Chang’e-5 samples consisting in a new type of differentiated lunar basaltic rock” to “by adding 2.0 Gyr Chang’e-5 samples consisting of a new type of differentiated lunar basaltic rock”

Change “From the new maps, the molar or atom ratio of Mg/(Mg+Fe)” to “From the new maps, the molar or atomic ratio of Mg/(Mg+Fe)”

Change “A young mare basalts unit is determined...” to “A young mare basalt unit is determined...”

“This identifies critical potential sites to constrain the late lunar magmatic and thermal history.” – This sentence is rather vague. It could be reworked to be more specific. Are you referring to future sample return mission landing sites?

“Lunar samples collected by six Apollo and two Luna missions...” – There were three Luna sample return missions, not two

“Apollo and Luna missions only revealed the evolution of Moon 3.0 Gyr ago, missing the second half of

the history of the Moon” – This implies the Moon is 6 billion years old

Change “Lunar soil samples having young age” to “young lunar soil samples”

Change “Chang’e-5 samples carry information about young volcanic activity on the Moon and is an indispensable...” to “Chang’e-5 samples carry information about young volcanic activity on the Moon and are an indispensable...”

“The classical and improved abundance equations...” I’m not sure what you mean by classical and improved equations. It may be simpler to include the equation you used.

“However, only TiO₂ and FeO abundances can be derived for the constraint of absorption diagnostic feature in the VNIR bands.” This sentence should be reworked for clarity.

“Throughout of the existing global distribution maps, varying degrees of differences are present in chemical abundance even with the same samples” to “Throughout existing global distribution maps, varying degrees of differences are present in chemical abundance even with the same samples” . Also, what “samples” are you referring to? Are you referring to locations on the Moon or physical lunar samples that were used to calibrate the maps?

Change “i) it exists a wide gap between the number of sampling points and the amounts of values to be estimated.” to “i) a wide gap exists between the number of sampling points and the amount of values to be estimated.”

Maps of lunar surface chemical abundances

Change “It may be due to the fact that Chang’e-3 and Chang’e-5 are both on the young Eratosthenian lava flows” to “It may be due to the fact that Chang’e-3 and Chang’e-5 are both on young Eratosthenian lava flows”

Comparison with previous lunar chemical maps.

“The differences of oxide abundance on the global or lunar lithologic unit’s surfaces are very significant.”
– This sentence is incomplete. Are you referring to differences between your map and previous maps?

New ground truth in division of geologic units

Data pre-processing

Change “In this work, the MI reflectance data are with a resolution of 59 m/pixel after topography shadow correction” to “In this work, the MI reflectance data have a resolution of 59 m/pixel after topography shadow correction”.

General comments

Extended data table 1 needs to list the numbers of the samples used.

The manuscript does not discuss the limitations of the SiO₂ map, and leaves readers with a skewed understanding of the range of SiO₂ values on the Moon. The upper limit of the SiO₂ map in Fig. 1 is 48.6 wt. %. However, we know from both the sample collection and from other remote sensing data, that highly silicic regions >60 wt.% SiO₂ occur. The manuscript needs a discussion on these limitations of the map and how to interpret the SiO₂ map, particularly in regions known to be silicic (e.g., Aristarchus, Compton-Belkovich volcanic complex, ect.)

Reviewer(s) Comments:**Reviewer: 1**

This is an interesting work on Chang'e 5 samples, which were used for training a 6-layer 1D CNN as the inversion model to predict the chemical contents in the new lunar maps precisely. On SELENE (KAGUYA) multiband imager (MI) data, each pixel represents a local lunar surface region with eight spectral observations, and 1D CNN model is thus trained over this 1D data to predict six major oxide abundances corresponding to the spectral observation based on the features extracted by the feature extraction module. The work shows certain merit and improved the earlier results in terms of accuracy; however, I noticed the following issues especially on the implementation of the deep learning (prediction) module, which should be addressed properly on the next revision round.

Response:

Thank you very much for your encouragement and suggestions. We thank you for valuable comments, which led to significant improvements in the inversion model. The detailed changes are summarized as follows:

1. The Authors stated that the training data is limited, i.e., “there is a very small number of returned lunar samples.” I could not see this number, later on referred as N . What is the size of the train set? This is the reasoning the Authors have mentioned using a 1D CNN model, which in fact has 6 layers and 100s of neurons (channels). If N is really small (e.g., $N < 500$), even this network will easily “overfit”.

Response: Thank you for your comments. In this work, considering spectral and spatial characteristic of SELENE MI data, a total of 115 lunar soil samples acquired from 55 lunar sampling sites of Apollo, Luna, and Chang'e-5 landing regions (which deemed to be relatively geologically uniform) were selected to represent the ground truth of chemical abundances. The number of samples used in the training process is 55. The description has been included in the Methods section (Sample selection). The details of adopted lunar samples available with Supplementary Table 1 (Supplementary Information). In order to mitigate overfitting with this very small number of returned lunar samples, the following two strategies were considered. First, a 1D CNN-based inversion model was designed in this study in light of the fact that the 1D CNNs have fewer parameters with local connectivity and weight sharing, resulting in less overfitting problem in the case of limited samples [27]. Second, the leave-one-out cross-validation (LOOCV) is used during the training process to select the best-performing model by evaluating its performance on a single validation set, and an early-stopping strategy is designed to avoid overfitting due to continued training. For the experimental results (Supplementary Fig. 1), one can see that the 1D CNN inversion model proposed in this study achieved a low RMSE and high R^2 under the LOOCV setting, which indicates that the proposed inversion model properly manage overfitting.

The reason to use 1D CNN and description of strategy to mitigate overfitting problem have been added in the Methods section (Deep learning-based inversion model) (page 7).

[27] Serkan, K. et al. 1D convolutional neural networks and applications: A survey. Mechanical Systems and Signal Processing 151, 107398(2021).

2. On the other hand, many other network models exist that especially suit to limited training data and have a better learning performance than CNNs, e.g., Transformers, Operational Neural Networks (ONNs), Convolutional Support Estimator Networks (CSEs), etc., especially when considering the CNN model used.

Response: Thank you for your comments and suggestion. The research work aims to invert the abundance of major oxides from the spectral images by establishing the complicated relations between the oxide abundances and spectral albedo characteristics. It should be noted that the corresponding spectral albedo characteristics at a certain band of different oxides can be considered as local features. The 1D CNN with local connectivity property has the ability to extract local features. Meanwhile, the weight sharing mechanism in 1D CNN can reduce the number of parameters to alleviate overfitting. Therefore, the 1D CNN is naturally suitable for the tasks in this study.

Although there are some models for the case of limited training data, not all of them are applicable to the task in this study. The Transformer [A] treats each element in a sequence (e.g., a word in a sentence) as a token, which is encoded (using techniques such as word2vec) as a dense vector before being fed into the model. For spectral images, it is difficult to pre-encode a spectral value in the sequence as a vector. Thus, in our opinion the Transformer is not suited to this task. Convolutional Support Estimator Networks (CSEs) [B], [C] was proposed for image processing which includes two phases, i.e., a pre-trained large model (such as ResNet or DenseNet) to extract features from the image data and a 2D CNN model with fewer layers to perform the subsequent tasks based on the extracted features. However, this task takes 1D sequences as input. Therefore, the CSEs are also unsuitable for this study. For Operational Neural Networks (ONNs) [D] and Generalized Operational Perceptrons (GOPs) [65], [66], they are both new variants of MLPs. According to the suggestions of the Reviewer, Generalized Operational Perceptrons (GOPs) was chosen as the comparison method in the revised paper.

The comparative experiments and results have been given in the Methods section (Ablation and comparative experiments) (pages 7 and 8) and Supplementary Table 3, respectively.

[A] Ashish V., Noam S., Niki P. et al. Attention is all you need. Proceedings of the 31st International Conference on Neural Information Processing Systems. 6000–6010 (2017).

[B] M. Yamaç, M. Ahishali, A. Degerli, S. Kiranyaz, M. E. H. Chowdhury and M. Gabbouj. Convolutional Sparse Support Estimator-Based COVID-19 Recognition from X-Ray Images. IEEE Transactions on Neural Networks and Learning Systems 32, 1810-1820 (2021).

[C] M. Ahishali et al. Advance Warning Methodologies for COVID-19 Using Chest X-Ray Images. IEEE Access 9, 41052-41065 (2021).

[D] Kiranyaz, S., Ince, T., Iosifidis, A. et al. Operational neural networks. Neural Comput & Applic 32, 6645–6668 (2020).

[65] Serkan K., Turker I., Alexandros I. et al. Progressive Operational Perceptrons. Neurocomputing 224, 142-154 (2017).

[66] Tran, D.T., Kiranyaz, S., Gabbouj, M., Iosifidis, A. PyGOP: A Python library for Generalized Operational Perceptron algorithms. Knowledge-Based Systems 182, 104801 (2019).

3. The Authors used a "Vanilla CNN" model although there are recently much better CNN models with enhanced learning capabilities and superior receptive fields size, e.g., UNet, VGG, ResNet, Inception, etc., providing CNNs with pooling layers, self-attention layers, skip connections, etc. that can improve the performance. Please note that solely using stride=2 will not guarantee an enhanced receptive field size.

Response: Thank you for your comments and suggestion. The recently proposed better CNNs, such as UNet for image semantic segmentation, and VGG, ResNet and Inception models for image classification, are all used for 2D image tasks and are not applicable to the data form in this study, where the spectral signature of each pixel should be analyzed for unmixing without the misleading spatial context. From the experimental results of LOOCV, it can be seen that the designed 1D CNN inversion model has achieved satisfactory performance. Therefore, the skip connections and complex self-attention layers are not used in order to keep the simplicity of the model. For receptive field size, our model contains 5 convolution layers with 1×3 kernel size, even if stride = 2 is not taken into account, the model has 1×11 receptive field size, which is larger than the length of the spectrum (8 bands) and it can fully satisfy the task demand. In the designed model, a convolutional layer with stride = 2 is adopted instead of a pooling layer to achieve the purpose of enlarging the receptive field and aggregating information, while avoiding information loss caused by the pooling operation [E]. In addition, the pooling operation can be considered as a special case of convolution, e.g., an average pooling operation of size 1×2 is equivalent to a convolution layer with convolution kernel [0.5, 0.5]. In reference [F], it has been shown that a convolutional layer with stride = 2 can be used in place of the pooling layer in the network. Consequently, the receptive field size of the model can reach 1×21 when stride = 2 is used in the 2nd and the 4th convolution layers of the designed 1D CNN inversion model.

[E] Sabour, S., Frosst, N., Hinton, G.E. Dynamic Routing Between Capsules. Proceedings of the 31st International Conference on Neural Information Processing Systems. 3859-3869 (2017).

[F] Springenberg, J.T., Dosovitskiy, A. et al. Striving for Simplicity: The All Convolutional Net. Proceedings of the 3rd International Conference on Learning Representations. 1-14 (2015).

4. With their limited connection and weight sharing capabilities, CNNs are preferred especially when the data dimension is high, e.g., images with hundreds of pixels or 1D signals with a long data vector like 2048×1 . However, in this work the input 1D data has only a data vector of 8×1 and thus a fully connected network can rather be used such as MLPs or their new variants Generalized Operational Perceptrons (GOPs), or even ELMs. Such networks can learn the interrelations of these 8 elements much better than 1D CNN with only 3×1 kernels.

Response: Thank you for your comments. In view of the fact that the task in this study is to invert abundance of oxides from a 1D spectral sequences, the reflectance characteristics exhibited by oxides in a certain band on the spectral profile can be regarded as local features. Therefore, the 1D CNNs with small-window local connectivity property is more suitable for the task. Meanwhile, the weight-sharing mechanism enables fewer parameters compared to the models such as MLPs, which makes it more effective for the case of limited sample data. Also, with comment 3, the 1D CNN inversion model designed has a 1×21 receptive field by stacking five convolutional layers with 1×3 kernel size, which is sufficient for learning the

inter-relationships of the 8 elements.

To further demonstrate the effectiveness of the proposed model, we compare the designed 1D CNN model with ELMs [64] and Generalized Operational Perceptrons (GOPs) [65],[66] in the Methods section (Ablation and comparative experiments) (pages 7 and 8). From the comparative results, the 1D CNN inversion model proposed in this study achieves the best LOOCV performance (Supplementary Table 3).

[64] Huang G. B., Zhu Q. Y., Siew C K. Extreme learning machine: Theory and applications. *Neurocomputing* 70, 489-501 (2006).

[65] Serkan K., Turker I., Alexandros I. et al. Progressive Operational Perceptrons. *Neurocomputing* 224, 142-154 (2017).

[66] Tran, D.T., Kiranyaz, S., Gabbouj, M., Iosifidis, A. PyGOP: A Python library for Generalized Operational Perceptron algorithms. *Knowledge-Based Systems* 182, 104801 (2019).

5. In the light of (1-4), the Authors should justify the choice of the network model, configuration and hyper-parameters. An ablation study is a must. In fact, they can investigate which network type and model with what configuration and parameters is the best to use in this problem. This will be very valuable.

Response: Thank you for your suggestion. In the revised paper, the ablation experiments of the proposed 1D CNN inversion model are performed on the model size and hyperparameters (i.e., learning rate and weight decay). With the same network structure (5 convolution blocks), the model size was changed by adjusting the number of channels in each convolution layer, and four model sizes were examined in the ablation experiments, i.e., 'small': [16, 16, 32, 32, 64], 'medium': [32, 32, 64, 64, 128], 'large': [64, 64, 128, 128, 256], 'huge': [128, 128, 256, 256, 512]. The two hyperparameters, i.e., the learning rate and the weight decay were set to {0.1, 0.01, 0.001, 0.0001} and {0.01, 0.001, 0.0001, 0.00001}, respectively. The ablation experimental results are presented in the Supplementary Fig. 5 and Supplementary Data 2. From the experimental results, the 'large' 1D CNN model with the learning rate = 0.01 and weight decay = 0.0001 achieves the best LOOCV performance.

6. The choice of the network model during training is missing. For each fold, the Authors are performing maximum of 100 epochs (and why 100?, did they notice a saturation in the training?) and as far as I see, there is no validation set. So how did you choose the model during training for the evaluation? At the end of the 100th epoch (when training is complete)? If so, how do you guarantee that this is the model that generalizes the best on the test set? I highly recommend to use a validation set within the train set (e.g, 20% of the train set can be spared for validation set) and then choose the CNN model in a particular epoch that yields the highest accuracy on the validation set. You will surely improve the accuracy further.

Response: Considering the few returned lunar samples, leave-one-out cross-validation (for a dataset with N samples, which can be viewed as N-fold cross-validation) [24] or K-fold cross-validation (K is generally not less than 10) [23] is a customary technique to evaluate the performance of designed inversion model. In this study, leave-one-out cross-validation was employed to avoid randomness in the sample division process as well as to maximize the utilization of the limited samples instead of dividing 20% of the samples as the validation set.

There are two stages for model choice. First, leave-one-out cross-validation (for a dataset with N samples, which can be viewed as N -fold cross-validation) was employed to train the 1D CNN inversion model. For each fold, one sample was selected as the validation set, and the remaining samples were used to train the model. During the training process of each fold, the evaluation results on the validation set are used to select the best model corresponding to this fold, while an early-stopping strategy based on the evaluation results is also utilized to avoid the problem of overfitting due to continued training. Thus, N models can be obtained in the first stage. For the adoption of the early-stopping strategy, the training process in each fold is accomplished within 100 epochs during the experiments, thus the number of epochs is set to 100. In the second stage, inspired by the work of [61] and [62], a simple but effective strategy, i.e., averaging the weights of the obtained N models is adopted to derive the final model for predicting the oxide abundance on the lunar surface. These N averaged models are the best models corresponding to each fold in leave-one-out cross-validation. The weight averaging process can be regarded as a model-level ensemble strategy, which ensures that the final model has a good generalization ability.

In the revised paper, the model training and the model choice strategy are supplemented in the Methods section (Deep learning-based inversion model) (page 7) and the code of the inversion model is provided at <https://github.com/hszhaohs/DL-IM>.

[23] Zhang, L. Zhang X., Yang M. et al. New maps of major oxides and Mg # of the lunar surface from additional geochemical data of Chang'E-5 samples and KAGUYA multiband imager data. *Icarus* 397, 115505 (2023).

[24] Wang, X., Zhang, J & Ren, H. Lunar surface chemistry observed by the KAGUYA multiband imager. *Planetary and Space Science* 209, 105360 (2021).

[61] Izmailov, P., Podoprikin, D., Garipov, T. et al. Averaging Weights Leads to Wider Optima and Better Generalization. *Conference on Uncertainty in Artificial Intelligence (UAI)*. 2018.

[62] Wortsman, M., Ilharco, G., Gadre, S.Y. et al. Model soups: averaging weights of multiple fine-tuned models improves accuracy without increasing inference time. *Proceedings of the 39th International Conference on Machine Learning*. 23965-23998 (2022).

7. Finally, the Authors should combine deep learning discussion in a single section. Some network details are scattered all along the manuscript. For example, the last paragraph of “Comparison with previous lunar chemical maps” section can be moved into the “Deep learning inversion model” section. They can provide all the details and block diagrams for the network models used in the study. This will improve the manuscript significantly.

Response: Thank you for your suggestion. The paper has been reorganized according to the guide for submission to the Nature Communications. The deep learning discussion has been arranged in the Methods section (page 7). The description of invention model and the ablation and comparative experiments subsections in are presented in the Methods section (pages 7 and 8), respectively.

Reviewer: 2

Congratulations on the completion of such a large project. I have a handful of minor comments to help improve the flow of the manuscript. I also have a concern about the framing of the SiO₂ map, detailed below. I look forward to reading the completed manuscript – Dr. Sarah Valencia

Response:

Thank you very much for your encouragement and for the valuable and meticulous comments, which led to significant improvements in the results of lunar surface chemistry map. The detailed changes are summarized as follows:

Abstract

1. “However, this mapping could only calibrate chemical features before 3.0 Gyr, missing the critical second half of the history of the Moon.” – Saying “half” here implies a much older age than the Moon actually is.

Response: Thank you very much for your comment. The sentence has been revised to “However, this mapping could only calibrate chemical features before 3.0 Gyr, missing the critical late period of the Moon.” (page 1).

2. “Therefore, young lunar soil samples with different chemical characteristics are necessary for mapping surface chemistry comprehensively.” – Why is it necessary that the samples are both young and have different chemical characteristics?

Response: The reason has been added in the abstract of the revised paper. And the sentence has been revised to “Young lunar soil samples with different chemical characteristics carry information about recent volcanic activity that are necessary for mapping surface chemistry comprehensively.” (page 1).

3. Change “by adding 2.0 Gyr Chang’e-5 samples consisting in a new type of differentiated lunar basaltic rock” to “by adding 2.0 Gyr Chang’e-5 samples consisting of a new type of differentiated lunar basaltic rock”.

Response: Thank you for your comments. The sentence has been changed in the revised paper (page 1).

4. Change “From the new maps, the molar or atom ratio of Mg/(Mg+Fe)” to “From the new maps, the molar or atomic ratio of Mg/(Mg+Fe)”.

Response: Thank you for your comment. The sentence has been changed in the revised paper (page 1).

5. Change “A young mare basalts unit is determined...” to “A young mare basalt unit is determined...”.

Response: Thank you for your suggestion. The sentence has been changed in the revised paper (page 1).

6. “This identifies critical potential sites to constrain the late lunar magmatic and thermal history.” – This sentence is rather vague. It could be reworked to be more specific. Are you referring to future sample return mission landing sites?

Response: The sentence has been revised to “The identified young mare basalt unit can be as critical potential sites in future sample return mission to constrain the late lunar magmatic and thermal history.” (page 1).

7. “Lunar samples collected by six Apollo and two Luna missions...” – There were three Luna sample return missions, not two.

Response: Thank you for your remark. It has been corrected (page 1).

8. “Apollo and Luna missions only revealed the evolution of Moon 3.0 Gyr ago, missing the second half of the history of the Moon” – This implies the Moon is 6 billion years old.

Response: The sentence has been revised to “Apollo and Luna missions only revealed the evolution of Moon 3.0 Gyr ago, missing the critical late period of the Moon.” (page 1).

9. Change “Lunar soil samples having young age” to “young lunar soil samples”.

Response: Thank you for your comment. The sentence has been changed in the revised paper (page 1).

10. Change “Chang’e-5 samples carry information about young volcanic activity on the Moon and is an indispensable...” to “Chang’e-5 samples carry information about young volcanic activity on the Moon and are an indispensable...”

Response: The sentence has been changed in the revised paper (page 1).

11. “The classical and improved abundance equations...” I’m not sure what you mean by classical and improved equations. It may be simpler to include the equation you used.

Response: Thank you for your remark. All the “abundance equations” in the paper have been revised to “abundance algorithms” (page 1).

12. “However, only TiO₂ and FeO abundances can be derived for the constraint of absorption diagnostic feature in the VNIR bands.” This sentence should be reworked for clarity.

Response: Thank you for your comment. The sentence has been revised to “The abundance algorithms determined only the TiO₂ and FeO contents with the defined sensitive parameters by performing a coordinate rotation in UV/VIS ratio versus VIS reflectance space.” (page 1).

13. “Throughout of the existing global distribution maps, varying degrees of differences are present in chemical abundance even with the same samples” to “Throughout existing global distribution maps, varying degrees of differences are present in chemical abundance even with the same samples”. Also, what “samples” are you referring to? Are you referring to locations on the Moon or physical lunar samples that were used to calibrate the maps?

Response: Thank you for your comment. The sentence has been revised to “Throughout existing global distribution maps, varying degrees of differences are present in chemical abundance even with the same lunar samples.”. And the “samples” is referred to the physical lunar samples that were used to calibrate the maps (page 1).

14. Change “i) it exists a wide gap between the number of sampling points and the amounts of

values to be estimated.” to “i) a wide gap exists between the number of sampling points and the amount of values to be estimated.”

Response: Thank you for your comment. The sentence has been changed in the revised paper (page 1).

Maps of lunar surface chemical abundances

15. Change “It may be due to the fact that Chang’e-3 and Chang’e-5 are both on the young Eratosthenian lava flows” to “It may be due to the fact that Chang’e-3 and Chang’e-5 are both on young Eratosthenian lava flows”.

Response: Thank you for your comment. The sentence has been changed in the revised paper (page 4).

Comparison with previous lunar chemical maps.

16. “The differences of oxide abundance on the global or lunar lithologic unit’s surfaces are very significant.” – This sentence is incomplete. Are you referring to differences between your map and previous maps?

Response: Thank you for your comment. The sentence has been revised to “The oxide abundances on the global or lunar lithologic unit’s surfaces between the new map and the previous maps show varying degrees of difference.” (page 5).

Data pre-processing

17. Change “In this work, the MI reflectance data are with a resolution of 59 m/pixel after topography shadow correction” to “In this work, the MI reflectance data have a resolution of 59 m/pixel after topography shadow correction”.

Response: Thank you for your comment. The sentence has been changed in the revised paper (page 7).

General comments

18. Extended data table 1 needs to list the numbers of the samples used.

Response: Thank you for your comment. The numbers of the samples used have been list in the Supplementary Table 1. (Supplementary Information)

19. The manuscript does not discuss the limitations of the SiO₂ map, and leaves readers with a skewed understanding of the range of SiO₂ values on the Moon. The upper limit of the SiO₂ map in Fig. 1 is 48.6 wt. %. However, we know from both the sample collection and from other remote sensing data, that highly silicic regions >60 wt.% SiO₂ occur. The manuscript needs a discussion on these limitations of the map and how to interpret the SiO₂ map, particularly in regions known to be silicic (e.g., Aristarchus, Compton-Belkovich volcanic complex, ect.)

Response: Thank you very much for this important suggestion.

Frist, there are gross differences in sampling size, i.e., remote sensing pixels in SELENE MI data measure areas dozens of meters in size, whereas the returned samples are from areas typically much smaller than a square meter. Second, since the sensitivity depth of reflectance spectra is limited to the upper few millimeters, oxide estimates represent only the upper-most surface materials. Therefore, only lunar soil samples acquired from sites

deemed to be relatively geologically uniform are used for mapping the lunar surface chemistry. We notice that the lunar samples with SiO₂ content higher than 50 wt.%, such as the samples from Apollo14 (14160, 14310, 14321, 14001), Apollo15 (15007, 15405, 15382, 15386, 15417, 15434, 15465, 15947, 15966, 15969), Apollo16 (60016, 65095) and Apollo17 (72275, 77538), belong to the KREEP basalt, breccia, immiscible Si-rich glass, or Si-rich melt inclusions. Thus, the above samples are not considered to estimate the oxide abundance. Meanwhile, the previous lunar chemical maps based on Chang'e-1 IIM and Diviner CF also used the same sample selection strategy, and the range of SiO₂ contents are about 39 wt.% - 48.2 wt.% and 37.3 wt.% - 46.4 wt.%, respectively. The Lunar Prospector Gamma-Ray Spectrometer (GRS) maps show the SiO₂ contents between 36 wt.% and 47 wt.%.

For the silicic regions on the Moon, i.e., Hansteen Alpha, Gruithuisen Domes, Aristarchus Crater, Lassell Massif, Helmet, Apennine Bench, Mairan Domes, and Compton-Belkovich volcanic complex, the SiO₂ contents are 45.09 wt.%, 44.36 wt.%, 43.14 wt.% (crater rim is about 46%), 47.36 wt.%, 46.59 wt.%, 46.82 wt.%, 46.98 wt.% and 42.87 wt.%, respectively. The estimated SiO₂ contents in these silicic regions are relatively high compared with the other lunar surfaces. However, it should be noted that these abundances can only reflect the regional average values of the lunar surface materials for the complex geologic settings. Most of these silicic regions are located at the interchange of the lunar mare and highlands or in the lunar mare, such as Aristarchus craters where the Si-rich materials are confined to rim and ejecta. Higher resolution orbital images from future spacecraft missions, particularly returned samples from silicic regions, will allow for examination of silicic materials from the upper subsurface, which in many cases could reveal the original surface SiO₂ abundances before mare contamination.

The principle of lunar samples selection is added in the Methods section (Sample selection) (page 6). The limitations of the SiO₂ map have been analyzed in the Results and Discussion section (Comparison with previous lunar chemical maps) (page 6).

REVIEWERS' COMMENTS

Reviewer #1 (Remarks to the Author):

I'd like to thank the Authors for their detailed revision. Most of my comments were addressed successfully. The manuscript organization is really good now and the quality of technical writing has improved. However, as a minor revision comment, I'd like to add the following: I understand that with $N=55$, it is not feasible to approach this problem in a conventional way, such as dividing the dataset into reasonable size train/validation/test sets. Even with the 5-layer CNN they have used, overfitting is inevitable -simply because the network size exceeds the dataset size several times. Any network will surely learn from the train set -in fact we can call it "memorize" perhaps and there is no guarantee that it will generalize well. N-fold X-validation is one way to address this as the Authors have followed but averaging the N networks for the final model will also not guarantee a reasonable generalization – unless this network is tested over an independent test set, which does not exist so far. So, I have accepted most of the arguments that Authors have made. One correction though: you can still use UNet, ResNet, etc., instead of your vanilla 1D CNN since they also have 1D versions available. I, however, would not recommend this again due to the extremely limited train size. The only alternative way I can foresee in this case is to apply transfer learning as an alternative way the Authors have followed. However, I agree that the results are quite satisfactory. In that the manuscript in the current form can be accepted for publication.

Reviewer #2 (Remarks to the Author):

Congratulations on completing this work. This draft adequately addressed my concern about the SiO₂ interpretation from the previous draft. I have 3 minor comments on this draft.

1. In "New ground truth in division of geologic units"

"Therefore, the Chang'e 5 samples plays important role in revealing mineral and petrologic characteristics of the Moon and reestablishing the lunar magma ocean (LMO) model." - Change "samples plays" to "samples play"

2. In "Deep learning-based inversion model"

"The training was concluded with the following settings:" – change "concluded" to "done"

3. We've had a miscommunication on what I meant about putting the sample numbers into supplementary table 1. The table reads

"The chemical contents presented here are the average values of the measured lunar surface chemical abundances from the returned samples." So, to me it seems that these oxide values are those that have

been measured in the laboratory on returned samples. If that is the case, then the sample identification number is what I was looking for you to include. If these are sample analyses done by other authors, then appropriate citations are needed. If these are not actual sample analyses, the caption needs to clarify what these values are.

Warm regards,
Sarah Valencia

Reviewer(s) Comments:**Reviewer: 1**

I'd like to thank the Authors for their detailed revision. Most of my comments were addressed successfully. The manuscript organization is really good now and the quality of technical writing has improved. However, as a minor revision comment, I'd like to add the following: I understand that with $N=55$, it is not feasible to approach this problem in a conventional way, such as dividing the dataset into reasonable size train/validation/test sets. Even with the 5-layer CNN they have used, overfitting is inevitable -simply because the network size exceeds the dataset size several times. Any network will surely learn from the train set -in fact we can call it "memorize" perhaps and there is no guarantee that it will generalize well. N-fold X-validation is one way to address this as the Authors have followed but averaging the N networks for the final model will also not guarantee a reasonable generalization – unless this network is tested over an independent test set, which does not exist so far. So, I have accepted most of the arguments that Authors have made. One correction though: you can still use UNet, ResNet, etc., instead of your vanilla 1D CNN since they also have 1D versions available. I, however, would not recommend this again due to the extremely limited train size. The only alternative way I can foresee in this case is to apply transfer learning as an alternative way the Authors have followed. However, I agree that the results are quite satisfactory. In that the manuscript in the current form can be accepted for publication.

Response:

Thank you very much for your encouragement and suggestions. We thank you for the valuable comments, which significantly improved the quality of the paper and made the inversion model more convincing. We agree with your comment that the Unet and ResNet can be helpful in achieving better performance of the model, but the very small number of training samples limits a proper use of them. Since vanilla 1D CNN has achieved satisfactory results, UNet and ResNet were not adopted in our current work with the purpose of keeping the simplicity of the model. However, with the launch of the lunar exploration plans, more lunar return samples will be available in the near future. Thus, we plan to consider the use of UNet and ResNet in the future research works. Indeed, the future returned lunar samples can also be used to test the performance of the model, and more data can help to train a more robust inversion model. We also agree that transfer learning can be a promising way to address the issue of model generalization, we will consider this in our research.

Thank you again for your suggestions and recognition, and we are very excited that we have a consistent point of view on this research task.

Reviewer: 2

Congratulations on completing this work. This draft adequately addressed my concern about the SiO₂ interpretation from the previous draft. I have 3 minor comments on this draft.

Response:

Thank you very much for your encouragement and for the valuable and meticulous comments, which led to significant improvements in the results of lunar surface chemistry map. The detailed changes are summarized as follows:

1. In "New ground truth in division of geologic units"

"Therefore, the Chang'e 5 samples plays important role in revealing mineral and petrologic characteristics of the Moon and reestablishing the lunar magma ocean (LMO) model." - Change "samples plays" to "samples play"

Response: Thank you for your comment. The sentence has been changed in the revised paper (page 6).

2. In "Deep learning-based inversion model"

"The training was concluded with the following settings:" – change "concluded" to "done"

Response: Thank you for your comment. The sentence has been changed in the revised paper (page 7).

3. We've had a miscommunication on what I meant about putting the sample numbers into supplementary table 1. The table reads "The chemical contents presented here are the average values of the measured lunar surface chemical abundances from the returned samples." So, to me it seems that these oxide values are those that have been measured in the laboratory on returned samples. If that is the case, then the sample identification number is what I was looking for you to include. If these are sample analyses done by other authors, then appropriate citations are needed. If these are not actual sample analyses, the caption needs to clarify what these values are.

Response: Thank you very much for your comment. The sample numbers used with corresponding references have been added in the Supplementary Table 1. (Supplementary Information)